# Sustainable Management Strategies for Fruit Processing Byproducts for Biorefineries: A Review

Alfred Błaszczyk [1,*] , Sylwia Sady [1], Bogdan Pachołek [2], Dominika Jakubowska [3] ,
Mariola Grzybowska-Brzezińska [3] , Małgorzata Krzywonos [4] and Stanisław Popek [5]

[1] Department of Natural Science and Quality Assurance, Institute of Quality Science, Poznań University of Economics and Business, Al. Niepodległości 10, 61-875 Poznań, Poland; sylwia.sady@ue.poznan.pl

[2] Department of Product Management, Institute of Marketing, Poznań University of Economics and Business, Al. Niepodległości 10, 61-875 Poznań, Poland; bogdan.pacholek@ue.poznan.pl

[3] Department of Market and Consumption, Faculty of Economics, Olsztyn University of Warmia and Mazury, Plac Cieszyński 1, 10-957 Olsztyn, Poland; dominika.jakubowska@uwm.edu.pl (D.J.); margrzyb@uwm.edu.pl (M.G.-B.)

[4] Department of Process Management, Faculty of Management, Wroclaw University of Economics and Business, Komandorska 118/120, 53-345 Wroclaw, Poland; malgorzata.krzywonos@ue.wroc.pl

[5] Department of Food Product Quality, Cracow University of Economics, Sienkiewicza 5, 30-332 Cracow, Poland; popeks@uek.krakow.pl

*   Correspondence: alfred.blaszczyk@ue.poznan.pl

**Abstract:** The fruit processing industry generates enormous amounts of byproducts, which are primarily removed through landfill or incineration. However, these processes cause carbon dioxide and methane emissions and release dioxin into the environment. The management of fruit processing byproducts is important for reducing the amount of food waste that is sent to landfills and for developing strategies through the reuse of these products for valorization and economic added value. Fruit processing byproducts are rich sources of bioactive compounds and fermentable and nonfermentable sugars. Therefore, these materials are very attractive feedstocks for developing integrated multifeed biorefineries that coproduce a wide range of natural products and bioenergy. The studies presented here have shown sustainable strategies for managing fruit processing byproducts via a biorefinery approach to achieve full valorization via a circular economy. The full valorization project proposed five main phases, namely, pretreatment, extraction, dark or aerobic fermentation, anaerobic digestion, and post-treatment, as well as two additional pathways to generate additional bioelectricity. When choosing the appropriate directions for the presented concept, a technoeconomic analysis should be carried out, considering the type of biomass and its availability at the site of the biorefinery and throughout the year of production. Applying the proposed concept of biorefineries in closed-loop technology is a promising way to enhance economic efficiency and decrease environmental influence in accordance with sustainable development.

**Keywords:** biochar; bioethanol; biogas; dye-sensitized solar cells; fruit byproducts; fuel cells; sustainable strategies

## 1. Introduction

The fruit processing industry is one of the main producers of large amounts of waste. Byproducts such as pomace, peels, trimmings, stems, skins, bran, and seeds account for more than 50% of produced fresh fruits during fruit processing. The global production of fruit waste, which is generated only by the processing industry, is estimated to reach more than 190 million tons per year (Table 1). However, this amount of waste is much greater than that generated at all cycle stages, starting with agricultural production, industrial manufacture, processing, and distribution. In response to the need to minimize the impact of waste disposal, fruit byproducts are currently landfilled, incinerated, and composted.

However, these processes cause carbon dioxide and methane emissions and release dioxin into the environment. In addition, biochar causes the loss of valuable biomass and nutrients, as well as economic losses.

**Table 1.** Global production of selected fruits and possible amounts of byproducts from their processing in 2015–2020.

| Fruit | | Global Production ($10^6$ t) According to FAOSTAT Database | | | | | | Typical Losses and Waste (%) | Ref. | Potential Byproduct Amounts ($10^6$ t) |
|---|---|---|---|---|---|---|---|---|---|---|
| | | 2015 | 2016 | 2017 | 2018 | 2019 | 2020 | | | |
| banana | | 114.95 | 112.11 | 113.29 | 116.65 | 117.52 | 119.83 | 30 | [1,2] | 33.67–35.95 |
| apple | | 82.37 | 85.09 | 83.12 | 85.91 | 87.48 | 86.44 | 25 | [3] | 20.59–21.61 |
| grape | | 76.52 | 74.43 | 73.51 | 80.04 | 77.00 | 78.03 | 20 | [4] | 14.70–16.01 |
| orange | | 75.58 | 72.99 | 73.39 | 73.45 | 75.99 | 75.46 | 50–60 | [5,6] | 43.79–45.59 |
| mango | | 46.79 | 47.07 | 52.00 | 53.51 | 55.03 | 54.83 | 60 | [7] | 28.07–33.02 |
| tangerine | | 33.16 | 32.24 | 32.65 | 34.16 | 38.56 | 38.6 | 50 | [8] | 16.12–19.30 |
| melon | | 25.71 | 26.62 | 26.70 | 27.1 | 27.01 | 28.47 | 30 | [9] | 7.71–8.13 |
| pineapple | | 25.81 | 25.95 | 27.39 | 28.33 | 28.21 | 27.82 | 30–60 | [1] | 15.94–17.00 |
| lemon | | 16.99 | 17.08 | 17.67 | 19.66 | 20.11 | 21.35 | 50 | [8] | 8.50–10.67 |
| grapefruit | | 8.88 | 8.99 | 8.66 | 9.04 | 9.26 | 9.34 | 50 | [8] | 4.33–4.67 |

For this reason, the European Landfill Framework Directive (1999/31/EC) and the Waste Framework Directive (2008/98/EC) have obliged EU Member States to minimize the number of biodegradable residues going to landfills. There are ongoing changes to this and other Framework Directives to set more restrictive limits on the landfilling of food waste. Therefore, managing fruit processing byproducts is important for reducing the amount of food waste going to landfills and developing reuse strategies for valorization and economic added value. In an effort to reduce global food waste, fruit processing byproducts have emerged as promising and sustainable feedstocks for bioenergy and biofuels, because they are rich in fermentable sugars and carbohydrates, including cellulose, starch, hemi-

cellulose, and pectin [10]. However, producing bioenergy or biofuels from fruit waste is not economically attractive [11]. Hence, there is a strategy to use integrated biorefineries as a technological way to convert fruit byproducts into various products, including biofuels, bioelectricity, organic fertilizers, and value-added products (Figure 1). This co-production of bioenergy and bioactive compounds improves access to renewable energy and bioproducts, promotes diversification, and creates jobs. Recently, many reviews have been published on the use of fruit byproducts as alternative sources of nutrients [12–14]. The nutritional characteristics of fruit byproducts can be found in repositories such as "Feedipedia" (http://www.feedipedia.org/ accessed on 6 February 2024) and "Feed Tables" (https://www.feedtables.com/ accessed on 6 February 2024).

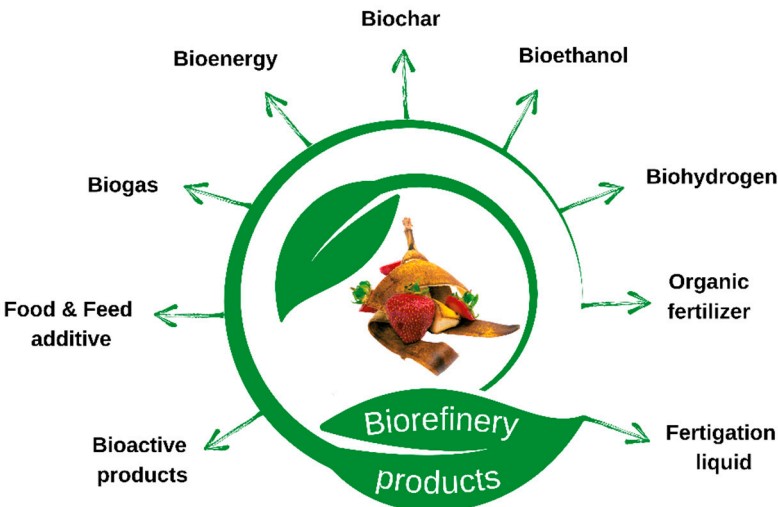

**Figure 1.** The main strategies for using fruit byproducts.

In the literature, there are many sustainable strategies for the valorization of fruit waste, mainly through the use of a biorefinery approach to produce valuable substances, bioproducts, biofuels, biofertilizers, and bioenergy [4,15,16]. However, some biorefinery models focus on recovering valuable substances or biofuel production. In addition, they are often associated with the partial valorization of fruit byproducts. Biorefining of citrus fruits or apple pomace focuses mainly on recovering valuable substances [17,18]. For example, researchers optimized the valorization of apple pomace to obtain valuable substances such as pectin and polyphenols [19]. Researchers developed other methods to partially valorize apple pomace, isolating pectin, monosaccharides, and cellulose-rich substances [20]. Developing a partial recovery process for passion fruit waste resulted in the recovery of sugars and furanic compounds, enabling the utilization of these wastes in the biorefining concept [21]. Pineapple waste was limited to valorizing natural substances such as antioxidants and organic acids [22]. Furthermore, researchers developed a sustainable biorefinery concept to extract pectin-enriched material and produce lactic acid from kinnow peel waste [23]. On the other hand, Molinuevo-Salces et al. mainly presented biofuel production during the biorefining of apple pomace [24]. Also, the conversion of apple pomace into biobutanol was developed [25].

This approach is incompatible with the closed-loop economy framework and is often not economically feasible due to the low efficiency of bioprocesses. Recently, some studies reported strategies for the extended valorization of apple waste to recover valuable products such as bioactive substances and biofuels [26]. For example, researchers developed a valorization process for pomegranate peels, which resulted in the recovery of pectin and polyphenols during extraction. In addition, the fermentation process yielded bioethanol from the extraction residue [27]. However, Borujeni et al. created a biorefinery that can turn apple pomace into high-quality pectin and lignin, as well as produce bioethanol at the same time. The socioeconomic aspects of apple pomace-based biorefinery were analyzed [28].

Other studies have developed the conversion of apple pomace into bioethanol, biogas, and bioproducts such as pectin, chitin/chitosan, and mycoproteins [27,29]. Also, a new concept of grape pomace biorefining was developed, resulting in biogas and biofertilizers [30]. Molinuevo-Salces et al. converted apple pomace into bioethanol and further subjected the postfermentation residue to anaerobic digestion to produce biogas [24]. In addition, Arun et al. developed the biorefining of pomegranate peels, focusing mainly on the recovery of bioactive substances and bioethanol production [31].

Recently, bibliometric research has shown that the main trend in managing fruit waste is the green recovery of high-value compounds. Other trends include the development of functional ingredients, compost, biofuel, and packaging materials [32].

Nowadays, several studies provide an overview of fruit byproduct processing trends [4,33]. Biorefinery concepts that can utilize fruit waste for the coproduction of bioenergy and biofuels with low volumes of value-added substances, particularly focusing on innovative biorefinery techniques, were reported [34,35]. However, they are often associated with the partial valorization of fruit byproducts. Therefore, there is a need for sustainable biorefinery approaches, which can provide full valorization of all types of fruit byproducts according to the circular economy.

This review reports the possible sustainable management strategies that are available in the scientific literature on fruit byproducts. For this purpose, we proposed a biorefining concept for the total valorization of fruit byproducts for a sustainable circular bioeconomy. The proposed biorefinery concept creates opportunities to produce bioelectricity to support biorefinery processes and high-value products. Finally, the most suitable solutions for sustainable management strategies for fruit processing byproducts are provided to company managers and other stakeholders.

This paper is organized as follows: Section 2 presents the method that was used to conduct this literature review; Section 3 is a description of the obtained results and a discussion of these. It is divided into two subsections: Section 3.1 describes the economic, social, and environmental determinants of the formation of fruit processing byproducts. Then, in Section 3.2, based on the literature analysis, the possibilities of implementing each technological stage of the proposed biorefinery concept are described. Section 4 provides conclusions, recommendations, and limitations.

## 2. Methodology

We have chosen the *narrative* (traditional) *review* [36–38] to summarize what has been written on the topic of fruit byproducts in relation to the economic, social, and environmental determinants of fruit processing byproducts and the integration of fruit byproducts in biorefineries.

To address the main purpose of this review, four phases—(1) design, (2) carrying out, (3) analysis, and (4) writing—were applied [39,40]. In the first phase, a research question was identified, and a search strategy for identifying relevant research was developed. We performed a comprehensive literature search through major academic databases, including Scopus, ScienceDirect, and Google Scholar. The bibliographic databases were searched for fruit byproduct-related fields, such as economic, social, and environmental determinants of fruit processing byproducts and for the integration of fruit byproducts in biorefineries. In phase two (conducting the research), regarding identifying articles for review, an exclusion step was performed to include only English language articles, as well as articles on the topic of fruit byproducts, including reviews, reports and papers, and articles that not only theoretically address the technologies of fruit byproducts in general but also develop visions and scenarios. The publication years ranged between 2000 and 2023. In the third phase, the analysis of the extracted articles was performed on the basis of the abstracts, and unrelated studies were excluded. In phase four, the results were used to construct the biorefinery concept for fruit byproducts, and these are presented in figures and tables, with descriptions and comments.

## 3. Results and Discussion

### 3.1. Determinants of Fruit Processing Byproducts

The fruit processing industry is one of the leading producers of byproducts. In fruit processing, byproducts such as pomace, peels, trimmings, stems, skins, bran, and seeds account for 20% to 60% of fresh fruits and contain large amounts of carbohydrates, dietary fiber, bioactive compounds, pectin, proteins, antioxidants, and phenolic compounds [41]. Table 1 shows the global production of selected fresh fruits and the possible amounts of byproducts that are generated from their processing.

Fruit waste is also caused by product damage during transport, storage, and processing. The increasing waste production has also led to the growing popularity of fruit juices, nectar, and frozen products in recent years [1].

The management of fruit processing byproducts faces challenges in transitioning from a linear to a circular economy [42]. The full utilization of fruit byproducts for producing bioenergy and a small volume of bioactive products has potential economic benefits. However, most valorization pathways that are discussed in the literature are limited to single raw materials [43–45]. Therefore, in this study, we designed a process scheme for the total valorization of fruit processing byproducts via a biorefinery approach for the coproduction of biofuels, bioenergy, biofertilizer, and bioactive products (Figure 2).

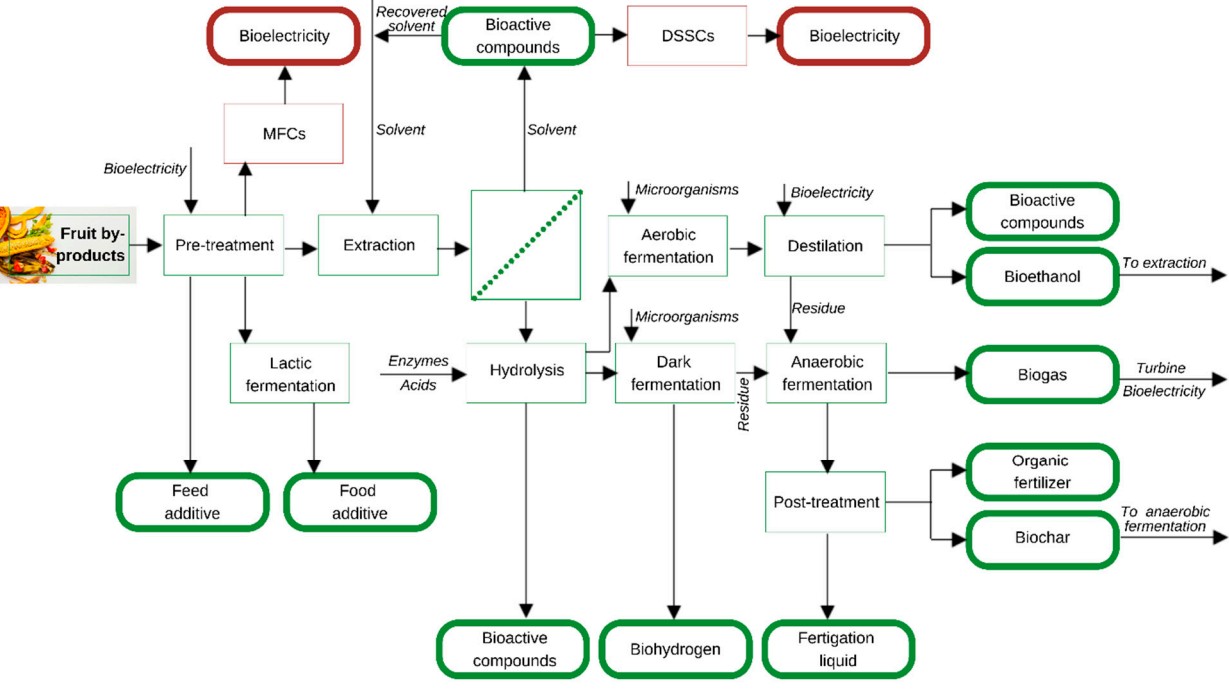

**Figure 2.** Concept of a process scheme for the total valorization of waste from fruit processing.

### 3.2. Integration of Fruit Byproducts in Biorefineries

The complete valorization project proposed five main technological phases, pretreatment, extraction, dark or aerobic fermentation, anaerobic digestion, post-treatment, and two other pathways to generate additional bioelectricity: microbial fuel cells and dye-sensitized solar cells (Figure 2). The current section describes the processes and their limitations, as well as the conditions under which fruit byproducts were processed in the literature.

#### 3.2.1. Pretreatment

The byproducts of fresh fruit processing are highly perishable due to their high moisture (approximately 70%) and sugar (approximately 50%) contents. Therefore, the biorefining approach for each fruit waste type begins with initial grinding. Waste then

undergoes a drying process to reduce the water content to approximately 5%, thus improving the shelf life of the raw material. The electricity for this process comes from biogas combustion in a gas engine with an electric generator or cogeneration units. The biggest problem with drying is that the bioactive compounds in fruit byproducts are sensitive to heat and oxygen. Different drying processes have been investigated for various fruit pomaces [46–48]. However, in recent years, unconventional drying methods, such as ultrasound, pulsed electric field, high-pressure, and various drying techniques, have been used to generate dried fruit pomace with increased nutritional parameters [49]. Several studies have examined the effects of different drying conditions on the degradation of bioactive compounds from fruit pomace [50].

Similarly, the number of volatile compounds, flavanols, phenolics, and anthocyanins and the higher DPPH radical scavenging activity in grape pomace were significantly lower in oven-dried samples than in freeze-dried samples [51]. The main disadvantage of the forced air and freeze-drying methods is the time needed. It takes 48–60 h to reduce the moisture content to approximately 5–6%. In addition, depending on the type of raw materials, the capacity of the plant and the duration of the cycle, freeze-drying is 4 to 8 times more costly than hot-air drying [46] and is economically feasible only for the drying of high-value fruit byproducts.

To fully utilize fruit byproducts, it is crucial to define drying conditions that maximize the retention of bioactive compounds while remaining economically feasible on a larger industrial scale. The dried biomass was then chemically and microbiologically analyzed for its contents of water, protein, carbohydrates, fat, dietary fiber, anthocyanins, polyphenols, microorganisms, pesticides, and heavy metals. Depending on the above parameters, dried biomass can be used directly as a feed additive, subjected to lactic acid fermentation to obtain a food additive, or subjected to the next step of the biorefining process, extraction.

### 3.2.2. Extraction

In the second technological stage of the designed process, bioethanol extraction is carried out to recover bioactive substances such as polyphenols, anthocyanins, carotenoids, essential oils, and pectin. During separation, the recovered ethanol is returned to the process. The electricity required for the above processes will come from biogas combustion.

Bioactive compounds can be extracted from fruit processing byproducts using conventional or nonconventional techniques. Conventional extraction, hydrodistillation, and maceration techniques are based on solvent and thermal extraction, which are time-consuming and energy-intensive, respectively, while simultaneously requiring a large volume of solvent, resulting in low selectivity and purity during extraction. They are also less suitable for heat-sensitive ingredients and are generally unsafe due to possible chemical contamination [52,53].

Other extraction methods exhibit shorter extraction times, high yields and selectivity, and lower solvent consumption. These techniques include ultrasound-assisted extraction, microwave-assisted extraction, supercritical fluid extraction, enzyme-assisted extraction, and pulsed electric field extraction [54,55]. Extraction methods that are assisted by green techniques, such as ultrasound extraction, can improve the extraction of heat-sensitive bioactive ingredients due to lower processing temperatures [56] and are more effective than conventional extraction [57]. Pingret et al. [58] reported that the polyphenol content in ultrasonically assisted apple pomace extract was 30% greater than that in conventionally extracted material. In addition, HPLC analysis showed that, unlike conventional methods, ultrasound does not degrade the polyphenols in the extracts.

Most of the above-mentioned methods are currently suitable for the extraction of polyphenols on a laboratory or pilot scale [50]. One of the critical points for the progress and greater use of these technologies at the industrial level is to optimize the conditions for their processing to achieve the most economical production. In the first step, the extraction process is optimized on a laboratory scale to achieve the highest quality and yield of the extracted components in the shortest possible time with a minimum amount of solvent residue

in the extracted components and minimum energy consumption. Next, the extraction process can be scaled up to technical and then industrial sizes and evaluated economically [59].

In summary, to fully utilize fruit byproducts, optimization of extraction methods and conditions that can maximize the recovery of bioactive compounds while remaining economically feasible on a larger industrial scale is crucial. The extraction residue is then sent to the third stage of the biorefining process, namely, dark or aerobic fermentation, or it can be directly passed through anaerobic digestion (Figure 2).

### 3.2.3. Dark or Aerobic Fermentation

The third technological stage begins with the pretreatment of the extraction residue, which leads to the hydrolysis of nonfermentable sugars using enzymes, bases, inorganic acids, or physical methods. The pretreatment methods used have a significant impact on the efficiency of further fermentation processes. After the hydrolysis of the biomass, fermentation inhibitors such as essential oils or polyphenols that are formed during this process are removed. Accordingly, to meet the needs of biorefineries, hydrolyzed biomass without inhibitors is then subjected to dark anaerobic fermentation to produce biohydrogen or to aerobic fermentation to obtain bioethanol for use as a solvent in the extraction process, or it can be directly transferred to stage 4 for anaerobic digestion (Figure 2).

In biological technologies, hydrogen can be generated via light-dependent processes, such as biophotolysis and photofermentation, or via light-independent processes, such as dark fermentation (DF) [60].

In DF, organic substances are converted to hydrogen by anaerobic bacteria in the absence of light and oxygen. Anaerobic bacteria such as *Clostridium*, *Enterobacter*, and *Bacillus* convert organic materials into hydrogen. In this process, byproducts such as alcohols and volatile organic acids are also formed [61]. Due to the low energy input and lack of oxygen generation, dark fermentation is a more promising technology for producing biohydrogen [62–65]. During dark fermentation, biomass is converted to a mixed gas containing $H_2$, $CO_2$, and other trace gases such as $CO$, $CH_4$, and $H_2S$ [66]. First, bacteria convert glucose to pyruvic acid, which is further converted to $H_2$ and $CO_2$ using pyruvate ferredoxin oxidoreductase and hydrogenase (Figure 3).

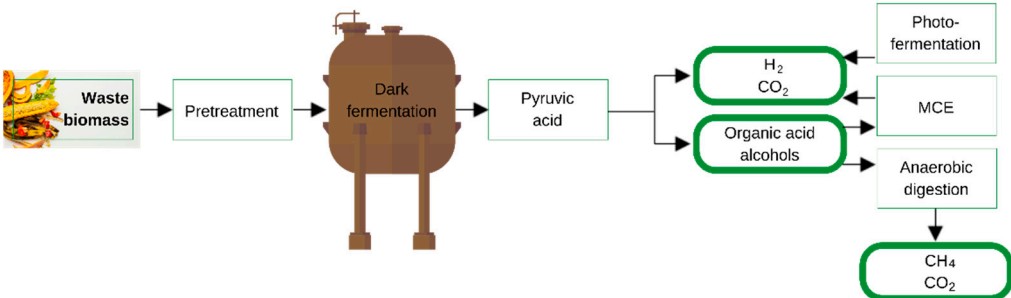

**Figure 3.** Integration of dark fermentation with other processes for bio-$H_2$ and bio-$CH_4$ production.

The maximum hydrogen production yield from the dark fermentation process is 4 mol $H_2$ per hexose molecule, which is 33% (based on sugar) [67]. Therefore, byproducts that are generated during dark fermentation are a solution for increasing the efficiency of the process. Waste biomass should be converted into bio-$H_2$ in two stages. The first stage is dark fermentation, which converts carbohydrates to hydrogen and organic acids/alcohols. In the second stage, the produced organic acids/alcohols are used in other processes to provide $H_2$ and $CH_4$ (Figure 3). The usual integrated mode is dark fermentation with anaerobic digestion and photofermentation [68,69].

Several key factors, such as the substrate, inoculum, inorganic nutrients, pH, temperature, and operational conditions, affect biohydrogen production during dark fermentation processes [70]. At higher temperatures (70 °C), a higher bio-$H_2$ production yield of 4 mol per 1 mol of hexose was observed [71]. However, this leads to increased en-

ergy consumption and affects the production costs of bio-$H_2$. A pH of 5.5 is optimal for hydrogen production [70,72].

Many studies have focused specifically on biohydrogen production from food waste, while few studies have examined the production of bio-$H_2$ from fruit byproducts (Table 2) [73,74]. Feng et al. [75] investigated acid and base pretreatments of apple peels to produce bio-$H_2$ in anaerobic digestion using river sludge. The maximum cumulative bio-$H_2$ production per 1 g of total solids (TSs) was 41.28 mL without pretreatment, 76.68 mL with $H_2SO_4$ pretreatment, and 101.08 mL with ammonia pretreatment. Doi et al. [76] obtained approximately 90 mL $H_2$/g TS from apple pomace using rhizosphere microflora without pretreatment. Therefore, the rhizosphere microflora can be used as an option for building a compact system for hydrogen production from apple pomace without pretreatment.

**Table 2.** Fruit byproducts as substrates for biohydrogen production in dark fermentation.

| Substrate | Microorganisms | Pretreatment | Temperature [°C] | Bio-$H_2$ Production | Reference |
|---|---|---|---|---|---|
| apple peel | Microbial consortium | not applied<br>$H_2SO_4$ solution<br>$NH_3$ liquor | 37 | 41.28 mL/g TS [a]<br>76.68 mL/g TS [a]<br>101.08 mL/g TS [a] | [71] |
| apple pomace | Rize rhizosphere microflora | not applied | 35 | 2.28 mol $H_2$/mol hexose | [76] |
| citrus peel | Microbial consortium | not applied<br>Alkali solution<br>hydrothermolysis | 30 | 13.55 mmol/L<br>7.27 mmol/L<br>8.19 mmol/L | [77] |
| citrus peel | *Enterococcus casseliflavus* | not applied | 37 | 13.9 mmol/L<br>1.09 mmol/h | [78] |
| banana peel | Anaerobic sludge | not applied | 37 | 352.8 mL<br>2.0 mL/h | [79] |
| banana waste | *Bacillus* sp. | not applied | 37 | 71 mL/g<br>6.1 mL $H_2$/h | [80] |
| banana waste | Autochthonous bacteria consortium | not applied | 37 | 70.19 mL/g<br>12.43 mL $H_2$/h | [81] |
| pineapple waste | Municipal sewage sludge | HCl solution | 37 | 5920 mmol $H_2$/g COD [b]<br>745 mL/h/L | [82] |
| fruit waste | *Clostridium strain* BOH3 | microwave moist heat | 37 | 359.97 mL/g | [83] |
| fully ripened fruits: grape, apple, pear | Sewage sludge | heat | 35 | 2.2 mol $H_2$/mol glucose | [84] |
| apple pulp waste | *Sporolactobacillus*, *Clostridium*, *Coprothermobacter* | not applied | 37 | 73.59 mL/g VS [c] | [85] |

[a] TS—total solids; [b] COD—chemical oxygen demand, [c] VS—volatile solid.

Camargo et al. [77] compared the effects of alkaline and hydrothermolysis pretreatment on bio-$H_2$ production during dark fermentation of citrus peels. A high production of bio-$H_2$ (13.9 mmol/L) was demonstrated using citrus peel without pretreatment. It has also been shown that banana waste does not require any pretreatment for bio-$H_2$ generation during the DF process. According to Da Silva Mazareli et al. [76], banana, citrus, and apple waste are easily fermented by microorganisms without complex pretreatment processes. pH 7.0 and 37 °C were the most suitable conditions for simultaneously increasing the optimum yield (70.19 mL $H_2$) and rate (12.43 mL/h $H_2$). According to the above studies, banana, citrus, and apple waste are readily fermented by microorganisms without complex pretreatment processes. Apart from bio-$H_2$, an anaerobic process generates several interme-

diate compounds, such as acetic, propionic, butyric, and lactic acids and alcohols. Nathoa et al. applied banana peels in two-phase anaerobic fermentation to produce bio-$H_2$ (yield: 352.8 mL/g VS) and acetic acid, propionic acid, and ethanol [75].

To summarize, by using appropriate microorganisms, it is possible to carry out DF without pretreatment, which will increase the economic efficiency of the process. Otherwise, the conditions for the DF process should be optimized for each type of fruit byproduct separately or for a mixture of fruit wastes with a constant composition. Unfortunately, hydrogen production by DF leads to a negative net energy balance [86]. Therefore, two-stage fermentation processes have been proposed to increase the energetic efficiency of fruit byproducts. In the first stage, biohydrogen is produced in DF, and in the second stage, methane is produced by mesophilic anaerobic fermentation. This two-stage process for simultaneous hydrogen and methane production is currently being developed and used in our field. The residue after DF, which contains organic acids and alcohol, is transferred to the next stage of this biorefining process, which is anaerobic fermentation (Figure 2).

Bioethanol is produced from fruit byproducts in the following three steps: biomass pretreatment, hydrolysis, and sugar fermentation (Figure 4). Orange, lemon, apple, lime, tangerine, and grapefruit peels are often used as biomass. These fruit byproducts consist of fermentable sugars such as fructose, glucose, and sucrose and insoluble polysaccharides such as hemicellulose, cellulose, and pectin.

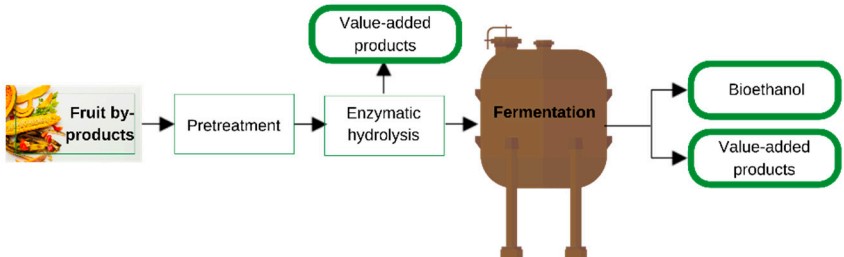

**Figure 4.** General steps in producing bioethanol and value-added substances from fruit byproducts.

The main disadvantage of the high production costs of bioethanol production from fruit byproducts is the pretreatment [87]. This production step aims to modify lignin to increase bioethanol production. The most commonly used pretreatment methods include mechanical and physicochemical techniques such as grinding, steam explosion, and acidic heating. Fruit byproducts such as citrus waste have a low lignin content and do not require harsh pretreatment before enzymatic hydrolysis, as is needed for lignocellulosic biomass. Therefore, these biomasses do not pose problems in terms of fermentation inhibitors such as sugar dehydration derivatives (furfural) and lignin degradation products (phenolic substances) that are generated under severe pretreatment conditions. For example, with lemon, tangerine, and orange peels, a short steam explosion requires 74% less energy than acid/base pretreatment [88], which is required as a pretreatment stage. In the case of apple pomace, which contains a large amount of lignin, an energy-intensive pretreatment, such as dilute acid or alkali treatment, is required before enzyme hydrolysis [83]. Magyar et al. [89] demonstrated that alkaline pretreatment is the most efficient method for producing bioethanol from apple pomace.

In the second step of bioethanol production, fermentable sugars are formed from lignocellulosic biomass, which consists of cellulose, hemicellulose, pectin, and lignin, through enzymatic or acid hydrolysis. Cellulose and hemicellulose are made of fermentable sugars such as xylose and glucose. Pectin consists of galacturonic acid and L-arabinose, which are not fermented by industrial microorganisms. Lignin is a phenolic polymer that interferes with enzymatic hydrolysis and can adsorb enzymes. High enzyme loadings, such as those of pectinase, cellulase, and glucosidase, are required to overcome the problems that are associated with pectin and lignin. High enzyme loads and the high costs of the enzymes that are used affect the economics of bioethanol production. Therefore, to reduce costs,

various additives, such as inexpensive soluble soy protein and internal enzymes, have been used in the literature [83]. Another inhibitor of the fermentation process is D-limonene, which is present in citrus waste. Therefore, Choi et al. [90] developed a new technique to remove and recover D-limonene after enzymatic hydrolysis, resulting in 12-fold greater bioethanol production.

The last stage of bioethanol production is fermentation, which is carried out mainly by industrial microorganisms such as *Saccharomyces cerevisiae*. In general, the first two stages of bioethanol production can be performed by separate hydrolysis and fermentation (SHF), simultaneous saccharification and fermentation (SSF), or separate hydrolysis and fermentation with vacuum evaporation (SHFE). Due to the high content of free sugars in fruit byproducts and the decreased formation of fermentation inhibitors, the productivity of bioethanol is much greater (1.1–4.7 g/L/h) than that of lignocellulosic biomass (0.1–0.9 g/L/h) [91]. Protzko et al. [92] showed that genetically engineered yeast can metabolize fruit byproducts to produce bioethanol and mucic acid. Many studies have focused on bioethanol production from fruit byproducts (Table 3).

**Table 3.** Fruit byproducts as substrates for bioethanol production.

| Substrate | Pretreatment | Enzymes | Fermentation Process | Microorganism | Ethanol Productivity [g/L/h] | Reference |
|---|---|---|---|---|---|---|
| orange peel | milling | pectinase, cellulase, glucosidase | SHF | *S. cerevisiae* | 4.7 | [93] |
| orange peel | acidic steam explosion | pectinase, cellulase, glucosidase | SSF | *S. cerevisiae* | 2.7 | [94,95] |
| orange peel | steam explosion | pectinase, cellulase, glucosidase | SSF | *Kluyveromyces marxianus* | 3.45 | [96] |
| lemon peel | steam explosion | pectinase, cellulase, glucosidase | SSF | *S. cerevisiae* | 67.8 [a] | [97] |
| tangerine peel | steam explosion | pectinase, cellulase, glucosidase | SSF | *S. cerevisiae* | 59.3 [a] | [85] |
| tangerine peel | popping | pectinase, cellulase, glucosidase | SHEF | *S. cerevisiae* | 46.2 [a] | [85] |
| tangerine peel | - | in-house enzymes | SSF | *S. cerevisiae* | 3.28 | [98] |
| grapefruit peel | - | in-house enzymes | SSF | *S. cerevisiae* | 2.40 | [93] |
| lemon peel | - | in-house enzymes | SSF | *S. cerevisiae* | 2.18 | [93] |
| apple pomace | acidic heating | cellulase | SHF | *S. cerevisiae* | 1.10 | [83] |
| apple pomace | alkali heating | pectinase, cellulase, hemicellulase | SSF | *S. cerevisiae* | 1.5 | [84] |
| apple pomace | acidic treatment | pectinase, cellulase, hemicellulase | SSF | *S. cerevisiae* | 190 g/kg DM [b] | [99] |

**Table 3.** *Cont.*

| Substrate | Pretreatment | Enzymes | Fermentation Process | Microorganism | Ethanol Productivity [g/L/h] | Reference |
|---|---|---|---|---|---|---|
| apple pomace | acidic treatment | pectinase, cellulase, hemicellulase | SHF | *S. cerevisiae* | 136,3 g/kg DM | [100] |
| apple pomace | ethanol treatment | pectinase, cellulase, hemicellulase | SHF | *S. cerevisiae* | 173,3 g/kg DM | [28] |

[a] Concentration of ethanol expressed in g of ethanol per g of 1000 kg of fresh substrate; [b] DM—dry matter.

Apple pomace has been identified as one of the most promising raw materials. This substrate can be fermented in the solid state with or without pretreatment. Vaez et al. [100] applied pretreatment of dried apple pomace with dilute sulfuric acid. Extraction of the liquid fraction gave pectin and residues, which after aerobic fermentation produced bioethanol. In addition, the solid fraction after the pretreatment process was subjected to anaerobic fermentation to produce biogas. The highest yield for 1 ton of dried apple pomace was 164 kg pectin, 99 L bioethanol, and 33.6 $m^3$ biogas. However, the maximum yield with enzymatic pretreatment was 190 g of ethanol/kg of apple pomace. The main disadvantage is the seasonal availability of this substrate [101].

As mentioned above, the main drawback of the high bioethanol production costs is the need for pretreatment and enzymatic hydrolysis. Pretreatment is an energy-intensive process. Enzymatic hydrolysis requires the use of expensive enzymes such as pectinase, cellulase, and glucosidase. There are several strategies to improve the economics of bioethanol production from fruit byproducts, such as removing fermentation inhibitors during the pretreatment process; applying in-house enzymes for enzymatic hydrolysis; coproducing other high-value bioactive products, such as essential oils, pectin, succinic acid, and polyphenols; and using the residue for biogas production [102,103].

In summary, bioethanol and bioactive compounds can be produced during aerobic fermentation. However, this technological process without pretreatment is the most economically promising due to its energy and chemical costs. To improve economic viability, most of the work on bioethanol production from fruit byproducts has focused on the coproduction of high-value products. The obtained bioethanol in this technological stage is cleaned by distillation and returned to the extraction process (Figure 2). In addition, separated value-added products can be used in the food, pharmaceutical, or cosmetic industries.

### 3.2.4. Anaerobic Digestion

In technological stage 4, wet anaerobic fermentation is carried out from the residues of the dark fermentation and the distillation process or directly after hydrolysis (Figure 2). Organic acids in the fermented residue are converted into $CH_4$ and $CO_2$ via aceto- and methanogenesis. Several attempts have been described in the literature to increase the energy efficiency of organic biomass using two-stage fermentation processes. However, there are a few examples of fruit byproducts used in this process [94,104]. Fruit byproducts often contain an essential oil (D-limonene) on their surface, which can hinder biodegradation and inhibit some biological processes. However, after removal of D-limonene, codigestion with other substrates and pretreatment are possible [28]. As a pretreatment method, silage increases methane production without significantly increasing operating costs for orange waste. Under these conditions, the methane generation increased by 119% in terms of methane generation potential without silage pretreatment, obtaining biogas with 70% $CH_4$ [100]. Jung et al. [105] studied a two-stage dynamic membrane bioreactor system to produce $H_2$ and $CH_4$ from food waste under mesophilic conditions. The highest average $H_2$ production rate was 7.09 ± 0.42 L/L/d, while the highest $CH_4$ production rate was 0.99 ± 0.02 L/L/d. In another study, the cofermentation of garden/food waste was

assessed in a two-stage process that combines hyperthermophilic dark fermentation (DF) and mesophilic anaerobic digestion (AD) [106]. The predictable energy production from DF and AD was 0.5 and 24.4 MJ/kg, respectively.

A recent study has shown that biomethane production from the organic fraction of municipal solid waste is a feasible energy resource that can meet sustainable production requirements [107].

Technologies based on an integrated, sustainable biorefinery approach to biogas coproduction were developed after prior extraction of valuable compounds, such as pectin, polyphenols, succinic acid, lactic acid, citric acid, and essential oils. Anaerobic digestion of organic waste leads to solid or liquid residues and biogas, a mixture of gases, mainly methane (50–70%). Biogas is burned in a combined heat and power unit to generate electricity and heat to supply biorefinery processes, and the remainder is transferred to the final technological stage (Figure 2).

### 3.2.5. Post-Treatment

The last stage of the biorefinery system is post-treatment, which is the critical aspect that is often neglected. In this process, postfermentation material is processed, which is estimated to constitute 0.2–0.47 tons of input from anaerobic fermentation [108]. The average water content in the digestate is 70–80% [109]. The digestate, which is rich in nitrogen, phosphorus, and organic matter, can be used as an organic fertilizer [110] or as a soil conditioner [111]. However, the digestate contains biodegradable organic residues and other contaminants that can pose a phytotoxic hazard. The digestate is usually separated into solid and liquid fractions by decanter centrifuges and screw press separators. In general, liquid digestate accounts for approximately 80% of the total mass of postdigestion material. The entire fermentation residue or the separated fractions can be used in different ways (Figure 5). The digestate can be used as a biofertilizer or in thermochemical processes, as well as for the production of value-added products and as a growing media for different microorganism cultures.

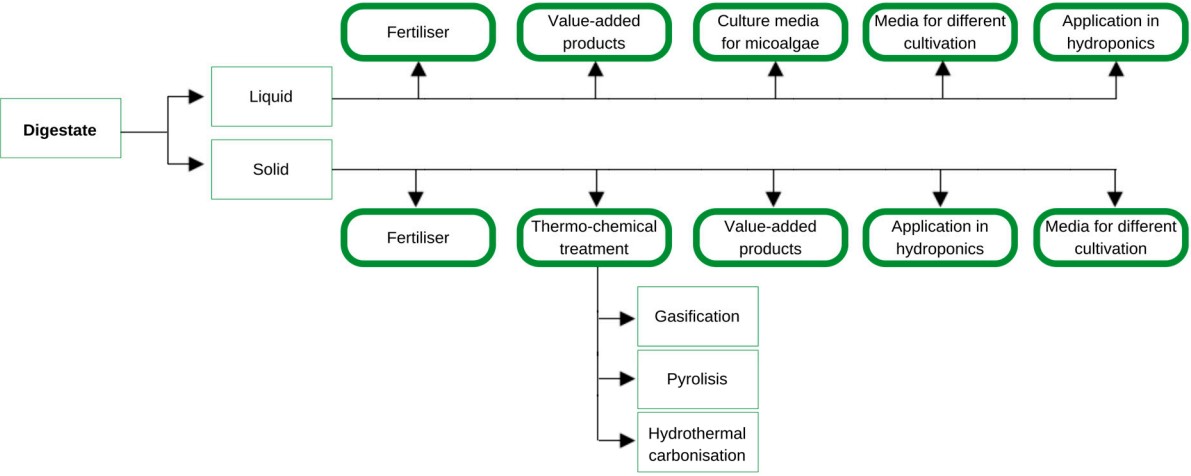

**Figure 5.** Possible strategies for accessing anaerobic digestate.

The liquid fraction can be used directly for agriculture. However, due to the high water content, transportation will determine the profitability. Therefore, several technologies, such as reverse osmosis, ammonia removal, membrane separation, and evaporation, have been developed to concentrate nutrients in liquid digestate. The above technologies can concentrate up to 67% of the feedstock nitrogen and consume less than 10% of the energy that is generated during anaerobic digestion [112]. The concentrated fraction can serve as a liquid biofertilizer for agricultural purposes. After additional modifications, the liquid digestate fraction can be used to cultivate microalgae [113].

There are several strategies for using a solid fraction of digestate in renewable and value-added materials, such as the following:

✓ Composting in biofertilizer [107];
✓ Pyrolysis in biochar production [114];
✓ Hydrothermal carbonization into fuel;
✓ Solid-state fermentation into hydrolytic enzymes, biosurfactants, and biopesticides.

The solid fraction of the digestate can be used directly as a biofertilizer for agricultural purposes. Sanitization of this organic fertilizer is not needed, which is recommended for digestate from animal products [115]. However, the application of digestate as fertilizer could increase $NH_3$ emissions and cause environmental problems such as acidification and eutrophication [116]. In addition, a large amount of unstable organic matter in the solid fraction of digestate can induce the activity of soil microbes, resulting in a loss of approximately 60–70% of the total nitrogen. Therefore, composting was applied before land application [117]. Composting of the solid fraction with bulking agents has been used to regulate the moisture content, adjust the C/N ratio, and accelerate the biodegradation rate [118].

Pyrolysis of solid digestate could be an attractive option for biochar production. The dried solid digestate was heated in an oxygen-limited process at 300–900 °C. The chemical composition of the digestate had a decisive influence on the properties of the biochar and its application. The potential of using tropical fruit waste to produce biochar was analyzed [119].

Hydrothermal carbonization of solid digestate could offer advantages in terms of energy efficiency and gas emission compared to pyrolysis. The process can be carried out in water at relatively moderate temperatures ranging from 180 to 250 °C, which means that the solid digestate cannot be dried. Therefore, hydrothermal carbonization is the most promising method for converting solid fermentation residue into valuable hydrochar and nutrient-rich process water. Hydrochar can potentially be applied as a soil amendment or as a solid fuel. Recent studies have shown that hydrochar has a positive impact on plant growth, soil properties, and the soil's microbial composition [120]. Other studies have shown that hydrochar can be used as a solid fuel and has a higher combustion reactivity than lignite [121]. Solid-state fermentation of solid digestate is an attractive option for producing bioproducts such as biosurfactants, biopesticides, and hydrolytic enzymes [122,123].

In summary, depending on the market demand and the economics of the technological stage, it is possible to apply postfermentation material to fertilizer and organic liquids or to produce biochar, biofuels, or bioproducts.

### 3.2.6. Alternative Stages in the Biorefining Process

The proposed concept of a full valorization of waste from fruit processing provides two alternative options for generating bioelectricity to support biorefinery processes. Both stages are marked in red in the scheme (Figure 2). In the first option, after pretreatment, fruit byproducts can be used directly to produce microbial fuel cells, which are a source of bioenergy. The second option is postextraction anthocyanins for manufacturing dye-sensitized solar cells.

### Microbial Fuel Cells (MFCs)

MFCs are bioelectrochemical devices that generate electricity from organic waste using electrons that are generated from biochemical reactions that are catalyzed by microorganisms [124].

The power generated by MFCs is affected by various factors, such as the type of microorganism, substrate, membrane type, electrode material, alkalinity, salinity, pH, and C/N ratio. Wang et al. [125] showed that the type of microorganism can determine the rate of waste decomposition, which affects the electrical energy. MFCs generate more electrical energy from substrates that are rich in different sugars than from monosaccharides [126]. MFCs perform optimally when the waste's C/N ratio is 30:1. The use of solid waste, such

as fruit byproducts, tends to increase the chemical oxygen demand (COD), which should be optimized to result in high power densities. In single-chamber MFCs, the electrode distance affects the amount of electrical energy. A shorter distance between electrodes generates more electricity due to the active surface area supporting a larger electrical gradient, as protons can more easily move to the cathode [127]. The pH plays an important role in the performance of microorganisms in MFCs. The optimal pH is in the range of 6–8. Another critical point of MFCs is the water content on the substrate. According to Wang et al. [128], a 40–60% moisture content in waste allows the process to occur under good conditions.

In the studies conducted by Miran et al. [129], an orange peel was applied as a substrate in a dual-chamber MFC, which generated a power density of approx. 350 mW/cm$^2$ with a current density of 847 mA/cm$^2$. However, the highest power density was achieved using papaya waste as fuel in single-chamber MFCs, at approximately 900 mW/cm$^2$ (Table 4).

**Table 4.** Performance of microbial fuel cells (MFCs) based on fruit byproducts.

| Substrate | MFC | Microorganisms | Maximum Voltage [V] | Current Density [mA/cm$^2$] | Power Density [mW/cm$^2$] | Reference |
|---|---|---|---|---|---|---|
| banana peel | dual-chamber | indigenous microorganisms | 0.492 | - | - | [130] |
| grape waste | single-chamber | no data | 0.5 | - | 825 | [131] |
| orange peel | dual-chamber | *Anaerobic sludge*: *Enterococcus*, *Paludibacter*, *Pseudomonas* | 0.59 | 847 | 358.8 | [123] |
| fruit waste | dual-chamber | fermentative bacteria | 0.26 | 1.0 | 24.2 | [132] |

Since MFCs are power systems that are based on anaerobic microorganisms, they convert fruit byproducts into electricity with little sludge production. However, the maximum power generation is relatively small, with a maximum value of 0.072 MJ/kg of food waste [133]. Therefore, this technology is still available at the laboratory scale. Further research needs to be carried out to address several disadvantages, such as the low power density and high cost of electrodes and membranes, which are required for the commercialization of this technology.

Dye-Sensitized Solar Cells (DSSCs)

DSSCs are the third generation of solar cells, developed by O'Regan and Gratzel [134]. The device is based on a mesostructured thin film of a wideband-gap semiconductor oxide modified by dye molecules (sensitizers). The sensitizer in DSSCs plays a crucial role in achieving higher solar-energy–electricity conversion efficiency [135]. Natural sensitizers such as anthocyanins, chlorophylls, carotenoids, and flavonoids are ideal candidates for green solar cells, because they are nontoxic, biodegradable, and easily extracted from plants or even plant byproducts such as leaves, peels, or pomace by using simple alcohol or water extraction processes [136]. For this reason, research into nature-based DSSCs has been primarily devoted to selecting the plant to isolate the sensitizer and find the best efficiency. Fruit byproducts such as leaves, peels, and pomaces are known to contain high levels of chemicals and can be used as sensitizers in DSSCs (Table 5).

Currently, the efficiencies of these DSSCs are much lower than those of synthetic sensitizer-based DSSCs, ranging from 0.002% to 2.63% [137,138]. Therefore, this technology, based on natural sensitizers, is still available at the laboratory scale. Further research needs to be carried out to address some of the disadvantages, such as low efficiency and stability, which is required for the commercialization of this technology.

**Table 5.** Fruit byproducts as sensitizers in DSSCs.

| Substrate | Sensitizers | Efficiency [%] | Reference |
|---|---|---|---|
| banana peel | carotenoids/chlorophyll | 0.21 | [139] |
| tangerine peel | flavanone | 0.71 | [140] |
| pineapple peel | chlorophyll/flavonols | 0.002 | [141] |
| black chokeberry pomace | anthocyanins | 0.105 | [131] |
| pomegranate leaf | chlorophyll | 0.597 | [142] |
| mangosteen peel | α-magnostin, anthocyanins | 2.63 | [132] |

The use of waste phytochemicals as renewable energy sources has positive effects on the economy, agro-industrial processes, and the environment.

## 4. Conclusions

The studies presented in this review have demonstrated strategies for managing fruit processing byproducts via a biorefinery approach to achieve full valorization via a circular economy. Natural products such as anthocyanins, carotenoids, vitamins, flavones, or flavanones can be recovered from fruit waste. After removing natural substances from fruit byproducts, the residue is subjected to biochemical processes, leading to the generation of biofuel, bioenergy, and organic fertilizer. These residues can be promising sources of biohydrogen or bioethanol. However, there is still much work needed for the cost-effective production of biohydrogen and bioethanol. The cost of producing bio-$H_2$ and bio-EtOH is largely dependent on the cost of the feedstock and the efficiency of the process. Improvements in the bio-$H_2$ and bio-EtOH production yield from fruit byproducts are currently in the early stages of research. Although anaerobic digestion has been commercialized, incorporating this process into the proposed biorefinery scheme will be challenging due to the chemical composition of the feedstock. Biogas is burned in a cogeneration unit to generate electricity and heat to support biorefinery processes. Postfermentation materials can produce biofertilizers, biochar, biopesticides, biosurfactants, and media for microalgae cultivation.

This work relates directly to one of the United Nations Sustainable Development Goals, namely, SDG 12 (Responsible Consumption and Production) by addressing various facets of sustainable consumption and production. This includes waste reduction, resource efficiency, innovation, and consumer awareness, which are all facilitated through the implementation of a biorefinery approach for the management of byproducts that are generated in fruit processing.

By effectively managing fruit processing byproducts, this study aims to reduce the amount of waste that is sent to landfills. Through the adoption of a biorefinery approach, this paper supports the establishment of a circular economy that decreases food losses throughout the production and supply chains and converts waste into valuable commodities such as biofuels, organic fertilizers, and bioenergy (SDG 12.3—waste reduction and circular economy).

The complete valorization of byproducts that are generated in fruit processing means the utilization of these materials to create a diverse array of valuable products. These include natural substances for the food, cosmetic, and pharmaceutical industries, as well as biofuels and organic fertilizers (SDG 12.4—valorization of byproducts).

The efficiency and economic viability of the technological stages that are involved in fruit processing through the recovery of bioactive compounds at each phase and the incorporation of byproducts into the production of various goods can be achieved (SDG 12.5—waste reduction).

The presented biorefining concept can stimulate the creativity of scientists, producers, and consumers simultaneously to accomplish the following:

✓  Increasing the efficiency and economy of the proposed technological stages;
✓  Increasing the level of waste management in the fruit industry;
✓  The development and implementation of new technologies for increasing the management of fruit byproducts;
✓  Developing new products based on fruit byproducts;
✓  Increasing the awareness of products resulting from the proposed technological processes.

Future studies should improve the efficiency of the proposed technological directions, and a technoeconomic analysis should be carried out, taking into account the type of biomass and its availability at the biorefinery site and throughout the production year. There is a chance that in the near future, the limitations of fruit byproducts in the production of bioactive compounds, biohydrogen, and bioethanol could be overcome, and these products could be considered economically viable components for bioproducts and a renewable energy-based economy.

The limitation of this work is the subjective selection of information from primary articles, which lacks explicit criteria for inclusion. The criterion used might mean that we have ignored relevant publications that were not published in English and not in 2000–2023, or that our attention was limited to certain studies. A narrative review typically does not rely on a strict systematic structure, which can hinder the replication of the study and does not guarantee a comprehensive review of the available literature. Narrative reviews are important to fill gaps in a field of knowledge, but they are subjective when determining the criteria for choosing the papers to be analyzed.

**Author Contributions:** A.B.: conceptualization, visualization, and writing; S.S.: conceptualization, and visualization; B.P.: conceptualization; D.J.: writing; M.G.-B.: writing; M.K.: writing; S.P.: writing. All authors have read and agreed to the published version of the manuscript.

**Funding:** Funding for this review was provided by the interuniversity research grants uep-uew-uek.

**Conflicts of Interest:** The authors declare no conflicts of interest.

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
