# Peer review of "Sustainable Management Strategies for Fruit Processing Byproducts for Biorefineries: A Review"

_sustainability, doi:10.3390/su16051717_

Round 1

Reviewer 1 Report

Comments and Suggestions for Authors

In the manuscript “Sustainable management strategies for fruit processing by-products for biorefinery: a review” the authors analyzed the possible strategies to use fruit processing by-products for their complete valorization according to a circular economy. This manuscript is well organized, and the drawn conclusions are coherent with the obtained results. The paper was well written!

Lines 34 – 35: Please, arrange the keywords alphabetically.

Lines 66 – 69: Please, explain in detail you hypothesis and predictions.

Comments on the Quality of English Language

The paper was well written!

Author Response

Thank you for your valuable feedback on our article.

1. Minor editing of English language required

Response: English language was corrected.

2. Lines 34 – 35: Please, arrange the keywords alphabetically.

Response: it was done.

3. Lines 66 – 69: Please, explain in detail your hypothesis and predictions.

Response: it was done. The following sentence was added:

The proposed biorefinery concept of using fruit by-products in the closed-loop technology creates opportunities to produce bioelectricity to support biorefinery processes and high-value products.

Reviewer 2 Report

Comments and Suggestions for Authors

The Manuscript is innovative, connected with the present researches in the most significant area of sustainability. The authors use appropriate references in the manuscript. The structure, methods and conclusions are well structured, present the main results obtained from the carried out research.

Suggestions toward the authors in order to improve the manuscript:

1. on row 463 there is something wrong (being are a rich ......) - to correct;

2. These should be more examples for the proper use of by-products obtained from fruit processing.;

3. What requirements should be met by the obtained by-products intended for the pharmacy, cosmetic, food industry, etc.?

Author Response

Thank you for your valuable feedback on our article.

Suggestions toward the authors in order to improve the manuscript:

  1. on row 463 there is something wrong (beingarea rich ......) - to correct;

Response: it was corrected.

  1. These should be more examples for the proper use of by-products obtained from fruit processing.;

Response: it was done.

  1. What requirements should be met by the obtained by-products intended for the pharmacy, cosmetic, food industry, etc.?

Response: the obtained by-products in biorefining, in order to be used in industry, must meet safety criteria in accordance with applicable law, such as all ingredients and raw materials used in the production of food, cosmetics and pharmaceutical products. For food, the basic regulatory act is Regulation (EC) No. 178/2002 of the European Parliament and of the Council of January 28, 2002; for cosmetics, Regulation (EC) No. 1223/2009 of the European Parliament and of the Council of November 30, 2009; and for pharmaceutical products, Directive 2001/83/EC of the European Parliament and of the Council of November 6, 2001 and Regulation (EC) No. 726/2004 of the European Parliament and of the Council of March 31, 2004.

Reviewer 3 Report

Comments and Suggestions for Authors

Dear authors, the paper is very interesting and I find several works that talk about by-products useful for the sustainable transition. I suggest some modifications:

- abstract report the main implications of your work

- section 1 is to be done. i don't see any literature review on the topic. i suggest some works https://doi.org/10.3390/agriculture13020286, https://doi.org/10.1016/j.spc.2022.10.014, https://doi.org/10.3390/su15107840

- section 1. you need to define the novelty of the work compared to existing and recent literature

- Sections 2 and 3 are very interesting but the approach used by the authors is not clear. create a methodology section and a results section

- with regard to the methodology indicate which methodology was used and what are its strengths. the choice of inputs was formulated on what basis?

- the results may be useful to compare against existing literature

- in the conclusions report the limitations of the work, the methodological and material implications. The technological choice on the recovery of these products plays a key role. What indications does this work provide? How does the social aspect support SDG 12?

Author Response

Thank you for your valuable feedback on our article. 

Dear authors, the paper is very interesting and I find several works that talk about by-products useful for the sustainable transition. I suggest some modifications:

1.  abstract report the main implications of your work

Response: It was done. Only one sentence was added because of limited number of words in abstract. The following part was added:

Application the proposed biorefinery concept in the closed-loop technology is a promising way to enhance economic efficiency and decrease the environmental influence according to sustainable development.

2. section 1 is to be done. i don't see any literature review on the topic. i suggest some works https://doi.org/10.3390/agriculture13020286, https://doi.org/10.1016/j.spc.2022.10.014, https://doi.org/10.3390/su15107840

Response: it was done. 

3. section 1. you need to define the novelty of the work compared to existing and recent literature

Response: it was done. The following part was added in Section 1:

“In the literature, there are many sustainable strategies for the valorisation of fruit waste mainly under a biorefinery approach to produce bio-products, biofuels, biofertilizers and bioenergy [5-6]. However, currently the proposed biorefineries only focuses on partial valorisation of fruit by-products, which is not completely compatible with the closed-loop economy framework and is not economically feasible due to the low-efficiency bioprocesses. Moreover, biorefineries focus only on processing one type of fruit by-product. In addition, they are focused on the recovery of either valuable substances or the biofuel production. For example, Kim et al. presented biorefining for citrus fruits, focusing mainly on the recovery of valuable substances [7]. Similarly, Dhillon et al. described the biorefining of apple by-product to produce high-value substances such as enzymes and organic acids [8]. In turn, Arun et al. presented the biorefining of pomegranate peels, focusing mainly on the recovery of bioactive substances and the bioethanol production [9]. Awasthi et al. described strategies for the valorization of apple waste to recover valuable products such as biofuels, biochemicals and bioactive substances [10]. However, Molinuevo-Salces et al. presented mainly the biofuel production during the biorefining of apple pomace [11]. On the other hand, Borujeni et al. developed the conversion of apple pomace into bioethanol and bioproducts such as pectin, chitin/chitosan and mycoproteins [12].Therefore, there is a need for sustainable conception in the biorefinery approach, which can provide a full valorisation of all types of fruit by-products according to the circular economy.

4. Sections 2 and 3 are very interesting but the approach used by the authors is not clear. create a methodology section and a results section

Response: we’ve changed the construction according to your suggestion. The methodology section and results section were added.

The current section 3 presents the processes and their efficiencies as well as the conditions in which fruit by-products was processed.

5. with regard to the methodology indicate which methodology was used and what are its strengths. the choice of inputs was formulated on what basis?

Response: below is a description of the methodology and its limitation.

"We have chosen the narrative (traditional) review [8,9] to summarize what has been written on the topic of fruit by-products in relation to economic, social, and environmental determinants of fruit processing by-products and integration of fruit by-products in biorefineries.

In order to address the main purpose of this review, four phases: (1) design, (2) conduct, (3) analysis, and (4) writing were applied [10,11]. In the first phase, a research question has been identified, and a search strategy for identifying relevant literature must be developed. We performed a comprehensive literature search through the major academic databases, including Scopus, ScienceDirect, and Google Scholar. The bibliographic databases were searched for fruit by-products related fields such as economic, social, and environmental determinants of fruit processing by-products, and integration of fruit by-products in biorefineries. In phase two (conducting the research) regarding identifying articles for review, an exclusion step was performed to include only English articles, articles on the topic of fruit by-products, including reviews, reports and papers, and articles not only theoretically address the technologies of fruit byproducts in general, but also developing visions and scenarios. Publication years were in range between 2000 and 2022. In third phase, the analysis of the extracted articles was performed on the basis of the abstract screening, not related were excluded. In phase four, results were used to build the biorefinery concept for fruit by-products and presented in figures and tables, with descriptions and comments.

Section 3.1 describes the economic, social and environmental determinants of the formation of fruit processing by-products. Then, in section 3.2, based on the literature analysis, the possibilities of implementing each technological stage of the proposed biorefinery concept were described.

The limitation of this work is the subjective selection of information from primary articles, which lacks explicit criteria for inclusion. The criteria used might cause that we have ignored relevant publications published not in English, and not in 2000-2022, or out attention was the limited attention paid to certain studies in order to make a point [12].

6. the results may be useful to compare against existing literature

Response: in the literature there are no biorefinery concepts for all types of fruit by-product based on based on closed-loop technology, which produces bioelectricity supporting the biorefinery and high-value products, including: anthocyanins, pectin, essential oil, polyphenols. The literature describes only biorefining for specific fruit wastes such as orange peels, berry pomace or apple peels and these biorefineries do not have completely closed loop.Therefore, in our manuscript we describe a new biorefinery concept for all types of fruit by-products with full valorization according to a circular economy.

7. in the conclusions report the limitations of the work, the methodological and material implications. The technological choice on the recovery of these products plays a key role. What indications does this work provide? How does the social aspect support SDG 12?

Response: we added the limitations of the work in the methodology. With regard to SDG 12 we added in the conclusions the following part:

"This work relates directly to one of the United Nations Sustainable Development Goals, namely to SDG 12 (Responsible Consumption and Production). The review discusses the biorefinery concept for fruit by-products which is directly related to significantly reduce the level of waste generation through prevention, reduction, recycling and reuse. Apart from that, this concept is indirectly related to the elimination poverty and hunger, rational use of natural resources, sustainable production and consumption and improvement of the quality of life."

Round 2

Reviewer 3 Report

Comments and Suggestions for Authors

Dear authors,

unfortunately this work has not improved its structure and its final message is very unclear.

- several responses to my comments are not implemented

- English issues

- structure of the work is not identified, you have only added parts

- the novelty of the work can be defined better than existing literature

Comments on the Quality of English Language

I suggest to check all text.

Author Response

- unfortunately this work has not improved its structure and its final message is very unclear.

Response:

The structure of the work was improved (look at the highlighted parts of the manuscript) and now the message from the work is clear. Section 3.1 describes the economic, social and environmental determinants of the formation of fruit processing by-products. Then, in section 3.2, based on the literature analysis, the possibilities of implementing each technological stage of the proposed biorefinery concept were described. After each technological step, a short summary regarding the status of that step was introduced.

In order to better understand the improvement made in the manuscript, I submit an additional version of the module with the changes.

Below is a short description:

The proposed concept consists of five main stages: pretreatment, extraction, dark or aerobic fermentation, anaerobic digestion, posttreatment, and two other pathways to generate additional bioelectricity: microbial fuel cells and dye-sensitized solar cells.

After the first technological stage (pretreatment) the dried biomass is chemically and microbiologically analyzed for the content of water, protein, carbohydrates, fat, dietary fiber, anthocyanins, polyphenols, microorganisms, pesticides, and heavy metals. Depending on the above parameters, it can be used directly as a feed additive, subjected to the lactic fermentation process to obtain a food additive, or proceed to the next step of the biorefining process, extraction.

In the second stage (extraction) to fully utilize fruit by-products, optimization of extraction methods and conditions that can maximize the recovery of bioactive compounds while remaining economically feasible on a larger industrial scale has to be applied. The extraction residue is sent to the third stage of the biorefining process, namely dark or aerobic fermentation, or it can be directly passed to anaerobic digestion.

In the next technological stage two-step fermentation processes have been proposed to increase the energetic efficiency of fruit by-products. In the first step, biohydrogen is produced in DF, and in the second step, methane is produced by mesophilic anaerobic fermentation. The residue after DF, which contains organic acids and alcohol, is transferred to the next stage of this biorefining process, which is anaerobic fermentation.

Instead of dark fermentation the extraction residue can be used in the aerobic fermentation to obtained bioethanol. After fermentation bioethanol is cleaned by distillation and returned to the extraction process. In addition, separated value-added products can be used in the food, pharmaceutical, or cosmetic industries.

The residue after dark or aerobic fermentation can be used in the next technological stage of this concept, namely in anaerobic digestion. Anaerobic digestion of organic waste leads to solid, liquid residues, and biogas. Biogas is burned in a combined heat and power unit to generate electricity and heat to supply biorefinery processes, and the rest is transferred to the final technological stage.

The final technological stage the post-fermentation material depending on the market demand and the economics of the technological stage, is possible to use into fertilizer and organic liquid or to produce biochar, biofuels or bioproducts.  

In order to support biorefining in terms of electricity, in our concept we have proposed additional paths for its production. In the first option, after pretreatment, fruit by-products can be used directly to produce microbial fuel cells which are a source of bioenergy. The second option is using post-extraction anthocyanins for manufacturing dye-sensitized solar cells.

- several responses to my comments are not implemented

Response:

We have responded to most of the suggestions. Only one was omitted, namely, “What indications does this work provide?” In the final version, it was added in the conclusion (highlighted part). See below:

The presented biorefining concept can be a factor stimulating the creativity of the simultaneous activity of scientists, producers, and consumers in order to:

  • increasing the efficiency and economy of the proposed technological stages,
  • improving waste management practices in the fruit industry
  • development and implementation of new technologies, increasing the management of fruit waste,
  • developing new products using by-products from the fruit industry,
  • increasing consumer awareness of products resulting from the proposed technological processes.

- English issues

Response:

The manuscript was checked and corrected by Curie from Springer Nature (AI – powered writing assistant to support researchers).

  • structure of the work is not identified, you have only added parts

Response:

Look at the highlighted parts of the manuscript to see how we improved the structure of the work. In accordance with your recommendations, we have described in detail the research methodology and its limitations and the structure of the work has been changed. In addition, the individual issues discussed in “Results and discussion” section are briefly described.

In order to better understand the improvement made in the manuscript, I submit an additional version of the module with the changes.

- the novelty of the work can be defined better than existing literature

Response:

The novelty of this manuscript lies in its idea of biorefining that can work with closed-system technology for all kinds of fruit by-products. This review reports the possible sustainable management strategies available in the scientific literature for fruit by-products. For this purpose, we proposed a biorefining concept for the total valorization of fruit by-products. We described the conditions and limitations of each technological stage based on literature data. The suggested biorefinery idea opens up chances to make bioelectricity to help biorefinery processes and valuable products like pectin, polyphenols, and anthocyanins. First, an analysis of the economic, social, and environmental determinants of the formation of fruit processing by-products was carried out. Next, we described the most recent valorization methods for the proposed biorefining concept. Additionally, we presented the most innovative techniques for recovering value-added substances from fruit by-products. Finally, we provide company managers and other stakeholders with the most suitable solutions for sustainable management strategies for fruit processing by-products.

Round 3

Reviewer 3 Report

Comments and Suggestions for Authors

Dear authors the work is not improved. However, I try to rewrite the limits of the work. I ask to indicate only the new changes in the revised version.

- Abstract contains 279 words, reduce it to 250

- fifth line you already talk about Table 1

- section 1 turns out to be weak in the literature. Have you checked bioenergy as an energy resource for achieving sustainable production within the papers published in 2023-2024? 

- Section 1. the novelty of the work goes from references 7 and 12. is ten lines enough to indicate the need for new scientific work? Reduce the text explaining the gap to be filled and how it will be addressed in this work

- don't start sentences too many times with authors' names.

- Figure 1. Is biomethane not an option?

- The methodological choice is justified with respect to work from 2013, 2015. It is not solid as an approach. Provide more recent references. What are the weaknesses of this approach? The limitation is certainly not the time frame and articles in English. 

- Also, you chose up to 2022 because it was subservient to other journal? By chance from another Publisher and therefore the certificate that was revised in English corresponds to this phase? Or the revised version?

- the comparison with the existing literature in the results you don't honestly understand what this paper says and what is called out by the authors based on the literature

- The conclusions are to be rewritten. Remove references, state the main implications of the work, and concretely explain how SDG 12 is achieved.

- Bioeconomy of sustainability: the role of by-products. I don't see this approach within this work and am very surprised

Comments on the Quality of English Language

Have you a copy with relative date related to English certificate?

Author Response

Dear Reviewer,

Below I have given answers to your comments:

- Abstract contains 279 words, reduce it to 250

Response:

It was reduced to 215 words.

- fifth line you already talk about Table 1

Response:

In this line we already mention table 1 because it contains data on the quantity of fruit by-products.

- section 1 turns out to be weak in the literature. Have you checked bioenergy as an energy resource for achieving sustainable production within the papers published in 2023-2024? 

Response:

We chose the literature study until 2022 because we started preparing the manuscript in mid-2023. Therefore, we could not write that the biorefining concept was based on research until 2023. However according to your suggestion, the manuscript was updated with new publications from 2023. References from 2024 were omitted because the methodology cannot state that the literature analysis was performed on the basis of data from 2000-2024. Apart from that, the references from 2024 do not change the proposed biorefining concept.

- Section 1. the novelty of the work goes from references 7 and 12. is ten lines enough to indicate the need for new scientific work? Reduce the text explaining the gap to be filled and how it will be addressed in this work

Response:

Due to the fact that biorefining concepts have recently been frequently published in literature, in the manuscript, we focused only on the description of the biorefining concept for fruit waste available in the literature because this is the subject of the manuscript.

The text explaining the gap was reduced. According to your suggestion the text was rewritten and more text was added regarding the novelty. Additional references were added as well.

- don't start sentences too many times with authors' names.

Response:

It was corrected.

- Figure 1. Is biomethane not an option?

Response:

Figure 1 contains biogas, which is the raw material for the production of biomethane. Of course, we can change it, but in this way, we should add all the products resulting from fruit waste.

- The methodological choice is justified with respect to work from 2013, 2015. It is not solid as an approach. Provide more recent references. What are the weaknesses of this approach? The limitation is certainly not the time frame and articles in English. 

Response:

The methodology is based on the publications of 2013 and 2015 as they constitute canons relating to the research methodology in the area of review work. Most scientists rely on these canons in preparing their research methodology. Therefore, we believe that these references should remain, but a new reference was added.

The weaknesses of this approach:

A narrative review typically does not rely on a strict systematic structure, which can hinder the replication of the study and does not guarantee a comprehensive review of the available literature. Narrative reviews are important to fill gaps in a field of knowledge, but they are subjective when determining the criteria for choosing the papers to be analyzed.

Authors are aware that we can overlook newer studies, especially those published after the completion of the review.

There is not always a clear research protocol specifying the steps of source analysis and selection, leading to a lack of transparency and difficulties in reproducing the study. However, we applied protocol which consisted of four phases: (1) design, (2) conduct, (3) analysis, and (4) writing. Future research should focus on systematic review, and clear definition of the protocol and inclusion and exclusions.

- Also, you chose up to 2022 because it was subservient to other journal? By chance from another Publisher and therefore the certificate that was revised in English corresponds to this phase? Or the revised version?

Response:

We chose the literature study until 2022 because we started preparing the manuscript in mid-2023. Therefore, we could not write that the biorefining concept was based on research until 2023. However, in the revised version we updated the literature and added some of the latest references.

- the comparison with the existing literature in the results you don't honestly understand what this paper says and what is called out by the authors based on the literature

Response:

Firstly, I do not know what paper the reviewer is talking about and secondly, I do not agree with the reviewer's opinion. All cited publications were reviewed and analyzed for their application to our proposed concept.

- The conclusions are to be rewritten. Remove references, state the main implications of the work, and concretely explain how SDG 12 is achieved.

Response:

The conclusions have been corrected in accordance with the comments. The reference 17 was removed. The state the main implications of the work was improved and SDG 12 was explained.

- Bioeconomy of sustainability: the role of by-products. I don't see this approach within this work and am very surprised

Response:

Of course, the manuscript fits into the bioeconomy of sustainable development through, among others, proposals for full valorization of fruit waste. By-products generated in all technological processes play a special role in this bioeconomy. At each stage of the proposed biorefining model, various by-products are created that will affect the bioeconomy of the technological processes carried out.

Therefore, the proposal of the biorefining concept developed as part of this work carries indications towards a positive impact on the environment and value for stakeholders, as well as the direction of changes in business models. Therefore, there is a clear need for organizations to adapt to sustainable development paradigms, because companies that are characterized by the so-called environmental responsibility are better perceived by both customers and business partners, and this attitude has a positive impact on the value they create, thus giving them a greater chance of defending themselves against a possible economic crisis.

I hope that the explanation was understandable and convincing.

With respect,

Alfred Błaszczyk

Round 4

Reviewer 3 Report

Comments and Suggestions for Authors

dear authors the paper is significantly improved. i suggest the latest changes to be implemented:

1. the novelty of the paper needs to be defined with respect to recent literature

2. eliminate all redundant elements in the text to allow the reader to read the paper with attention to the key concepts

3. report the limitations of the search

4. some references: i) https://doi.org/10.1016/j.spc.2022.10.014 ii) https://doi.org/10.3390/su15043679

5. I would make the conclusions more streamlined with a section on implications and future research directions in the previous section

Author Response

Dear Reviewer,

Below I have given answers to your comments:

  1. the novelty of the paper needs to be defined with respect to recent literature

Response:

The novelty of the manuscript was defined with respect to recent literature.

  1. eliminate all redundant elements in the text to allow the reader to read the paper with attention to the key concepts

Response:

Redundant elements in the text were removed.

  1. report the limitations of the search

Response:

The limitations of the search were reported at the end of manuscript.

  1. some references: i) https://doi.org/10.1016/j.spc.2022.10.014 ii) https://doi.org/10.3390/su15043679

Response:

Suggested references were cited in the manuscript.

  1. I would make the conclusions more streamlined with a section on implications and future research directions in the previous section

Response:

The conclusions were corrected.

I hope that the explanation was understandable and convincing.

With respect,

Alfred Błaszczyk